# The Relationship between Cellphone Usage on the Physical and Mental Wellbeing of University Students: A Cross-Sectional Study

**DOI:** 10.3390/ijerph19159352

**Published:** 2022-07-30

**Authors:** Muhammad Daniyal, Syed Fahad Javaid, Ali Hassan, Moien A. B. Khan

**Affiliations:** 1Department of Statistics, The Islamia University of Bahawalpur, Bahawalpur 63100, Pakistan; muhammad.daniyal@iub.edu.pk; 2Health and Wellness Research Group, Department of Psychiatry and Behavioral Sciences, College of Medicine and Health Sciences, United Arab Emirates University, Al-Ain P.O. Box 15551, United Arab Emirates; sjavaid@uaeu.ac.ae; 3Department of Media Studies, The Islamia University of Bahawalpur, Bahawalpur 63100, Pakistan; alihassan@iub.edu.pk; 4Health and Wellness Research Group, Department of Family Medicine, College of Medicine and Health Sciences, United Arab Emirates University, Al-Ain P.O. Box 15551, United Arab Emirates

**Keywords:** physical health, mental health, cell phone addiction, youths, Pakistan

## Abstract

(1) Background: The study aims to examine the use of cell phones on physical and mental health status and their impact on personality among university students. (2) Methods: A cross-sectional study was conducted using a semi-structured questionnaire. The association of physical and mental health variables with the demographic variables was examined using Pearson’s correlation and χ^2^-test. The binary logistic regression model was further used to predict the probabilities of negative impact on personality due to excessive use of cell phones. (3) Results: A total of 400 participants participated with a mean age of 24.45 ± 3.45 years. The average eye strain was more in High cell phone users HCPU than in LCPU and that difference was significant *p* = 0.000. The average neck pain was more in (HCPU) than Low cell phone users (LCPU) and there was a significant difference between the two groups with *p* = 0.006. The average weight gain was more in HCPU than LCPU and that difference was significant *p* = 0.000. Considering back pain, back pain was found more in HCPU as compared to LCPU with a statistical difference at *p* = 0.027. Cell phone usage significantly correlated with eye strain (r = 0.577, *p* = 0.000), neck pain (r = 0.543, *p* = 0.000), back pain (r = 0.611, *p* = 0.000), weight gain (r = 0.423, *p* = 0.000), depression (r = 0.430, *p* = 0.000), loneliness (r = −0.276, *p* = 0.002), and mood disorder (r = 0.608, *p* = 0.000). Eye strain, neck pain, and back pain was observed more in HCPU than in LCPU. HCPU felt they gained more weight when compared to the respondents in the LCPU group. HCPU felt more changes in mood and feeling low when compared to the LCPU, while LCPU group felt more lonelier when compared to the HCPU group. (4) Conclusions: The study highlights a significant association between excess use of cell phones and negative effects on physical and mental health wellbeing. Based on the results, it is recommended that more physical activities and alternative to minimize cell phone usage should be planned for the students. Public health policy makers and stakeholder need to address the ill effects of excessive use of cell phones through novel policies., especially young students, and alternatives to reduce their cell phone activities.

## 1. Introduction

Cell phone use is in excess, as it is one of the primary sources of information and communication, with more than 6.5 billion users worldwide [1]. Young adults spend more time on cell phones for social media, playing games, and other entertainments, as a means of communication or for academic purposes [2]. Excessive use of cell phones raises concerns about mental and physical health among young adults [3,4].

Though studies have demonstrated that the use of cell phones can have positive benefits in supporting physical and mental wellbeing [5], recent reviews have argued that excessive cell phone use is a type of addictive behavior and can affect wellbeing [6]. A three-year longitudinal study conducted among adolescents found using cell phones to be a significant predictor of depression in emerging young adults [7]. Furthermore, adults’ excessive cell phone usage was associated with depressive mood, anxiety, and loneliness [8,9]. Interestingly, users spending excessive time on a cell phone had higher stress levels [10]. It was noted in the study that problematic cell phone usage is associated with psychological distress and emotional dysregulation [11]. Excessive cell phone users were also found to be associated with Obsessive-Compulsive Disorder behaviors [12] and symptoms related to attention-deficit hyperactivity disorder (ADHD) [13]. Immense use of cell phones and an adverse attitude and feeling of anxiety can increase the risk of depression and anxiety [4,14]. Furthermore, several studies conducted among excessive cell phone users have validated the effects of problematic cell phone use, including digital stress [15], low self-esteem [6], worries and angers [16], loneliness [17], anxiety, depression and mood disorders [18,19].

Mobile phone addiction is rapidly gaining acceptance as a serious psychosocial condition. It has been identified as being regulated by the same brain circuits as in substance based and other behavioral addictions. While there is widespread concern that there is a close connection between social wellbeing and mobile phone usage, this link is less robust [20,21].

However, some evidence suggests that children who receive a phone earlier than others have difficulty adjusting to its use, and hence age is an essential mediating factor in the impact of mobile devices [20,21].

Excessive usage of cell phones was common during the night, leading to sleep disturbances leading to perceived physical ill-being, exhaustion and symptoms related to physical ill-health [22]. Furthermore, dependence on cell phones can have adverse effects on lifestyle, such as dietary habits and daily routines, which results in overweight [23,24,25]. Other studies have indicated a possible association between cell phone dependence and childhood obesity, pointing out that excessive usage of cell phones serves as a potential risk factor for obesity [26,27].

Musculoskeletal problems associated with cell phones ranged from 1% to 67.8% [28]. A systematic literature review reported a high frequency of musculoskeletal problems among adults related to the use of mobile devices ranging between 8–89% [29], which includes [30], back pain [31], and eye strains [32]. This is due to postural instability, poor exercise to associated muscles, excessive neck movements or hand overuse, and strain due to staring with the eye leading to musculoskeletal problems [33,34]. It is essential to evaluate the impact of the mental and physical wellbeing of the vulnerable young adults who are more prone to ill being and can have long-term negative effects leading to poor quality of life and excess sickness absences. Such negative effects can have a major economic impact due to an increase in disability-adjusted life years, causing a burden to the family and society.

There are differences in the degree of smartphone usage between different cultures, with China and Germany demonstrating more significant levels of smartphone usage and Scandinavian countries reporting lower levels of usage [13]. The adverse effects of smartphone usage have not been extensively in developing countries, but there appears to be limited evidence of significant levels of usage leading to several physical and mental health problems with associated affects like poor school performance [13,35].

Few studies have reviewed the mental wellbeing of Pakistani youths, however the studies were focusing more on cyber victimization [36]. Few studies have specifically looked at wrist pain [37], shoulder pain [38] and musculoskeletal problems in children due to cell phone [39]. To the best of our knowledge, no study comprehensively evaluates the impact of cell phone users’ hours, mental wellbeing, and physical wellbeing in Pakistan Hence we aim to investigate the effect of cell phone use on physical and mental wellbeing among Pakistani university students.

## 2. Material and Methods

### 2.1. Data Collection and Study Design

The estimated sample size required was calculated to be 400 respondents with a margin of error of 5% and a confidence level of 95% [40]. The cross-sectional study was conducted among university students between 1 October to 22 October 2021. Simple random sampling was used to recruit the samples in the study. A total of 400 respondents participated in the survey using google forms as a web-based questionnaire.

### 2.2. Inclusion and Exclusion Criteria

This cross-sectional study consisted of both male and female students. The subject selection criteria included candidates over 18 years who could understand English, fill up the online questionnaire, and consented to participate in the study. The exclusion criteria were any known medical conditions that could lead to neck or back pain and pre-diagnosed mental health.

### 2.3. Questionnaire Design

Demographic characteristics such as gender, age, education status, duration, daily smartphone usage time, and purpose of smartphone use in a typical day were studied. Information about physical and mental health outcome variables was computed from the previous validated questionnaires [41,42,43]. The physical health outcome variables were constructed for eye strain, neck pain, weight gain, and back pain associated with mobile phone usage. The response rate was divided into Yes or No response. The mental health outcomes included depression, loneliness, and mood disorder were from the previous validated questionnaires. UCLA loneliness scale was used to measure the feelings of loneliness variable. Respondents gave response to each question as either O (“I often feel this way”), S (“I sometimes feel this way”), (“I rarely feel this way”), N (“I never feel this way”). The PHQ-9 has been used as the depression scale, consisting for each question as “0” (not at all) to “3” (nearly every day). The Mood Disorder Questionnaire (MDQ), a screening scale for a mood disorder [41,42,43]. The cell frequency has been computed as the median hours of cell phone use by the participants of the study. On the basis of hours spent on the cell phones, the participants have been divided into two groups i.e., LCPU and HCPU. The participants who used cell phone below median were considered in the group of LCPU and those who used above median were considered in the HCPU group. The draft of the survey questionnaire was piloted initially on ten representative populations. Later minor adjustments were made to the survey questionnaire. Data was collected through an online survey questionnaire link which was sent through electronic and social media (Facebook, WhatsApp, and emails) to university students.

### 2.4. Ethical Considerations

This cross-sectional study was approved by the human ethical committee of The Islamia University of Bahawalpur (A52/12/06/2021) and is conducted per the Helsinki declaration for recruiting human subjects. Consent was obtained from every respondent after briefly explaining the purpose of the study in a separate section in the online form. We have followed Strengthening the Reporting of Observational Studies in Epidemiology (STROBE) guidelines for reporting our study [44].

### 2.5. Statistical Analysis

Cronbach’s alpha was computed to calculate the reliability of the questionnaire. Normality of the data was checked using the Kolmogorov–Smirnov test. Before applying the formal *t*-test, the assumption of the equality of variance was evaluated by Leven’s test. Descriptive statistics included mean age, median hours spent on mobile in a day, and percentage of responses were used to describe the demographic and mental and physical characteristics of the study participants. Chi-square analysis was performed to measure the strength and significance of the association between cell phone usage with physical and mental health variables. Binary logistic regression models were used to predict the probability of negative impact on personality by using excessive use of cell phones among the different states of physical and mental health issues. All analyses were performed by the statistical software package SPSS version 26. The results were selected as statistically significant if the computed *p* ≤ 0.05.

## 3. Results

The study showed an excellent reliability with a Cronbach’s alpha α = 0.798. Out of the 400 participants who participated 180 (45%) were females and 220 (55%) were males. The mean age of the respondents was 24.45 ± 3.45 years. 67.5% of the respondents were graduates, and 32.5%were undergraduates. The participants’ median daily time spent using cell phones was 6.16 ± 2.60 h per day (Table 1). The participants’ median daily time spent on cell phones was 6.16 ± 2.60 h per day.

The average age of the respondents differed significantly between LCPU and HCPU (*p* = 0.002). Considering gender, there was a significant difference between the two groups (*p* = 0.001). The average high cell phone users were found more in intermediate and pre-graduate educational status with a significant difference (p=0.001). The mean age differed significantly between the LCPU and HCPU (t = 1.798, *p* = 0.042). Significance difference was also observed in gender considering both the groups (t = 2.687, *p* = 0.001). The median mobile phone use was not observed significantly between LCPU and HCPU (t = −1.345, *p* = 0.179). We also observed a significant difference in mean educational status between LCPU and HCPU groups (t = 4.667, *p* = 0.001) The average eye strain was more in HCPU than in LCPU and that difference was significant *p* = 0.000. The average neck pain was more in HCPU than LCPU and there was a significant difference between the two groups with *p* = 0.006. The average weight gain was more in HCPU than LCPU and that difference was significant *p* = 0.000. Considering back pain, back pain was found more in HCPU as compared to LCPU with a statistical difference at *p* = 0.027. Considering the mental health issues, depression was found more in HCPU as compared to LCPU and that difference was statistically significant with *p* = 0.000 Loneliness and mood disorders were also found more in HCPU as compared to LCPU with a significant difference *p* = 0.000 for both.

The association of mental and physical health issues with cell phone use, the overall respondents have been divided into two groups i.e., LCPU and HCPU. Those who were equal to the median values were discarded and to remove the unbiasedness of the study the whole study was divided into an equal number of respondents in each group.

From Table 2, it can be observed that eye strain was observed more in HCPU as compared to LCPU with a significant association χ^2^ = 191.11, *p* = 0.021. Considering the neck pain, the HCPU declared that they feel neck pain more as compared to the LCPU group with a significant association χ^2^ = 175.23, *p* = 0.000. A significant association was observed between the weight gain and mobile phone usage and respondents in the group HCPU declared that they felt more weight gain as compared to the respondents in the LCPU group (χ^2^ = 187.14, *p* = 0.000). The same situation was observed between back pain and cell phone use. A slight significant association was observed between the neck pain and mobile phone use, and the group HCPU declared more pain in the back as compared to the LCPU group χ^2^ = 217.12, *p* = 0.037. Figure 1 shows the graphical representation of the proportion of physical health issues based on the LCPU and HCPU.

The association of cell phone use with mental and physical health issues was measured by χ^2^—A test with the *p*-values. In this current study, cell phone usage significantly correlated with eye strain (r = 0.577, *p* = 0.000), neck pain (r = 0.543, *p* = 0.000), back pain (r = 0.611, *p* = 0.000), weight gain (r = 0.423, *p* = 0.000), depression (r = 0.430, *p* = 0.000), loneliness (r = −0.276, *p* = 0.002), and mood disorder (r = 0.608, *p* = 0.000).

HCPU group felt more depressed when compared to the LCPU group (χ^2^ = 74.713, *p* = 0.000). Loneliness was also found associated with cell phone use but loneliness was found more in the LCPU group as compared to the HCPU group (χ^2^ = 16.935, *p* = 0.003). Mood disorder was also observed in the HCPU group of the respondents as compared to the LCPU group with a significant association (χ^2^ = 147.318, *p* = 0.001). Figure 2 shows the graphical representation of the proportion of mental health issues based on the LCPU and HCPU.

To measure the effect of physical and mental health factors affecting the overall personality, we have used binary logistic regression because the dependent variable which has been taken as Negative impact on personality (NIP) was categorical. The assumption of parallel lines was also tested which confirmed the use of this technique and justified its preference over simple regression models.
$$\theta \left(Y=k\;|\;X=x_{mi}\right)=logit\varphi \left(x\right)=\ln\left[\frac{\varphi \left(x\right)}{1-\varphi \left(x\right)}\right]=\beta_{ok+}\beta_{1k}x_{1i}+\ldots +\beta_{nk}x_{ni}$$
where “Y” denotes the vector of dependent variables, and “X” denotes the vector for independent variables. The number of observations is given by “i” and “m” denotes the number of independent variables. Table 3 represents the binary logistic regression models, which show the odds of negative impact on overall personality (NIP) under physical and mental health issues associated with excessive use of cell phones. Negative impact on overall personality (NIP) has been taken as the dependent variable with “1 = yes and 0 = No”. The question was asked at the last of the survey from the respondent “Do they feel a negative impact on their overall personality?” The respondents gave their opinion in either “Yes” or “No”. The respondents who are in the HCPU group associated with a high level of depression, loneliness, and mood disorder were included as the response for modeling purposes. The omnibus test of model coefficients is significant (χ^2^ = 201.880, *p* = 0.024) showing that the model with these explanatory variables performs better in predicting the outcome. The model’s *p*-value is statistically significant. The overall percentage is the percentage of time-predicted categories from the model which is 80.7 indicating the correctness of the model. Larger values give good results. The reference category of each independent variable was a lower-level category. Considering the physical health issues, respondents who have eye strain have higher odds of NIP as compared to those who do not have eye strain (β_1_ = 3.074, *p* = 0.000).

The result of the study showed that respondents who have neck pain have higher odds of NIP as compared to those who do not have neck pain (β_2_ = 2.583, *p* = 0.000). Students who have back pain have higher odds of NIP as compared to those who do not have back strain having the coefficients (β_3_ = 2.831, *p* = 0.011). It can be concluded from the study that respondents in the HCPU group who have gained weight have higher odds of NIP as compared to those who did not gain weight (β_4_ = 2.561, *p* = 0.001). As far as mental health issues are concerned, having depression among the respondents has higher odds of NIP than those who have a low level of depression (β_5_ = 2.266, *p* = 0.034). It was also noted in the study that those who feel loneliness and mood disorder have higher odds of NIP than those who have a lower level of these conditions with the coefficients (β_6_ = 1.295, *p* = 0.000) and (β_7_ = 1.716, *p* = 0.026) respectively.

## 4. Discussion

The study demonstrated a significant association between cell phone excess use and mental and physical health problems. The symptoms include increase in neck and back pain along with changes in mood and feeling depressed. To the best of our knowledge this is the first study to investigate the relationship between use of cell phones and physical and mental changes among Pakistani young adults.

In our current study, the average cell phone use by students was 6.16 ± 2.60 h per day. Over the last three decades, the world has seen a rapid transformation in ways we communicate, work and relax. Mobile devices have found there use in all these domains. The proliferation of mobile electronic devices has meant we now have 106 mobile cell subscriptions per 100 people globally [45]. Excessive mobile device usage, such as smartphones and tablets, has been found to have a negative impact on individual mental and physical health [29,46,47].

We observed higher numbers of eye strain, neck pain, back pain, and gain in weight in our study. The average eye strain, back pain, neck pain, and weight gain differed significantly between HCPU and LCPU groups. This is in line with emerging evidence that excessive mobile phone use is associated with headaches, fatigue, impaired concentration, memory disturbances, and poor sleeping patterns [48]. Studies have varied with the age groups that have received attention, with a particular focus upon young people [24,48,49,50]. Evidence suggests, excessive mobile use has been found to result in eye strain in users with the symptoms reported in 10 to 67 percent of the studied population [46,51]. This can be associated due to straining to use with the mobile device’s screen. Mustafaoglu and colleagues found that musculoskeletal, including neck, pain was significantly associated with participant’s mobile usage, specifically upper and lower back pain [29]. Interestingly, the study showed the pain varies according to how the device was used. A more active and varied use of mobile devices may mitigate the physiological effects associated with such devices.

Our study showed that weight gain differed significantly between LCPU and HCPU groups. Weight gain has an ambivalent association with mobile phone usage. In some cases, excessive usage has been found to increase the risk of obesity because mobile use displaces exercise. However, other studies have ascertained that excessive mobile use can be associated with positive benefits such as the use of fitness apps that can increase the level of physical exercise in users [49,50,52]. This reflects the degree to which the harm caused by mobile device usage may vary according to the kinds of uses of such devices, with positive uses potentially rebalancing negative effects.

The participants of the HCPU group were more likely to report loneliness, depression, and mood disorders with a statistically significant difference from the LCPU group. Our findings are supported by the previous studies [14,53], which it was observed that immense use of cell phones together with an adverse attitude and feeling of anxiety can increase the risk of depression and anxiety. Mobile device use has been found to have a negative effect on people’s mental health. In children, excessive mobile phone usage can result in compulsive buying, low mood, tension, anxiety, and leisure or boredom [47]. Excessive mobile phone usage among undergraduate students was predictive of higher levels of depression and anxiety, particularly where related to excessive social media usage in female students [54]. Overall, results consistently suggest that mental health issues can be affected by smartphone usage, but there are some variations on the ways in which these might have an impact according to age and gender [54]. Loneliness can be both a cause and effect of smartphone addiction. Loneliness can prompt mobile phone addiction and may also exacerbate the situation in sufferers [55]. Hence, the effects of smartphones can often serve as a proxy for social loneliness, with this a particular issue in recent years resulting from the social isolation caused by the Coronavirus pandemic lockdowns [56]. Our study found participants from the HCPU group as more likely to report depression. The causative links between these factors are not fully understood, but social isolation, reduced motivation, and social comparison where people compare their lives to the self-presentation of others on social media have been implicated [50]. Our findings are in line with the previous literature, that mobile phone addiction is associated with depression, especially in younger age groups [6,57]. There is a more general link between excessive mobile phone use and other mood disorders [57]. These conditions have been found to be co-morbid, in which fatigue, anxiety and depression can be linked as both cause and effect of the other; this suggests that excessive mobile phone usage can create a self-perpetuating link between usage and mood disorders [20,58].

Our results suggest that secondary to physical and mental health problems associated with HCPU, the overall personality of the participants was negatively affected. This was studied by binary logistic regression. The odds of negative impact on the personality in the presence of physical and mental health issues were also observed higher as compared to those who do not have these issues. These observations were found consistent with the previous studies [28,59]. The relationship between excessive mobile phone usage and personality traits has also received some attention in the literature. It has been suggested that problematic mobile phone usage can be associated with addictive personality, in that problematic mobile phone usage was more likely to be reported in cases where similar personality traits such as narcissism, neuroticism and impulsivity were identified [21,60].

The ways in which high levels of mobile phone usage can be alleviated has been considered through an examination of different interventions. Applications that limit the level of smartphone usage have been shown to have moderate success [61]. Exercise as a response to smartphone usage has also been shown to have a significant effect, in that it can be used to treat those with problematic usage [62]. Similarly, cognitive behavioral therapy has been shown to have some impact [62]. In extreme cases, the use of non-smartphone telephones might offer a potential response, but there is limited support for this intervention in the literature [63].

Based on the study findings, we suggest more physical activities for the students, especially young students, and alternatives to reduce their cell phone activities. The study has some limitations. The results of this study are based upon the data collected from the response of students selected from a public sector university of Pakistan and hence a limited generalizability Second, because we relied on self-reported information, there is a risk of social desirability bias. Finally, in this cross-sectional study design may not adequately reveal the causality of the factors. Nevertheless, this study was the first to investigate the association between cell phone usage on physical and mental health in Pakistan. More extensive multi-centric studies to determine the causality effect are proposed.

In conclusion, mobile devices have been implicated in a wide range of negative effects on individuals’ physical and mental wellbeing. However, this is dependent on the nature of use and may be balanced by the positive effects offered by such devices. Efforts should be made from stakeholders to assist student’s health and wellbeing through appropriate communication and policies. The relationship between physical and mental health is a serious global concern. Hence it is necessary for future research based on larger and broader samples from diverse age groups, to examine the associations between cell phone usage and the physical and psychological wellbeing of individuals.

## Figures and Tables

**Figure 1 ijerph-19-09352-f001:**
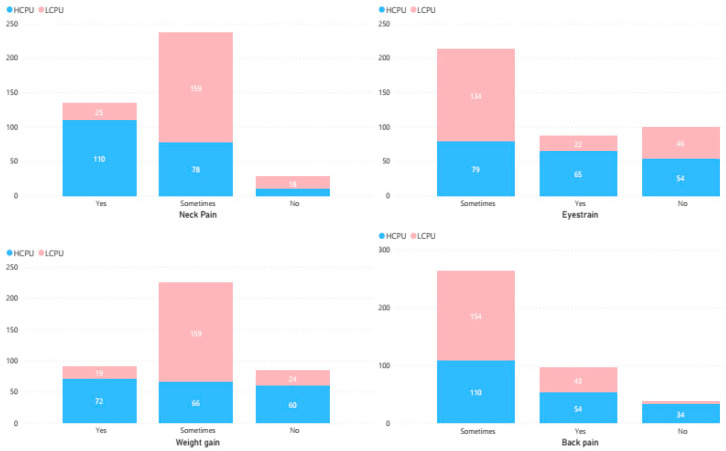
Proportion of neck pain, eye strain, weight gain and back pain based on LCPU and HCPU.

**Figure 2 ijerph-19-09352-f002:**
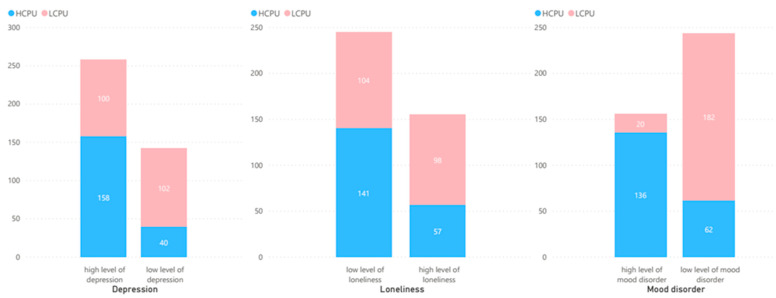
Proportion of depression, loneliness and mood disorder based on LCPU and HCPU.

**Table 1 ijerph-19-09352-t001:** Respondents’ general demographics and attributes with significance.

Attributes	Low Cell Phone Users (LCPU)	High Cell Phone Users (HCPU)	Test Statistic	*p*-Value
Age (years)	25.343 ± 6.74	24.437 ± 6.259	1.798	0.042 < 0.05 ***
Mean ± S.D.				
**Cell phone use (h in a day)**	5.06 ± 2.54	7.28 + 2.13	−1.345	0.004 < 0.05
**Median ± S.D.**				
**Gender (%)**			2.687	0.001 < 0.05
*Male*	110	98		
*Female*	92	100		
**Educational Status**			4.776	0.001 < 0.05
*Pre-graduate*	127	143		
*Postgraduate*	75	55		

*** significant if *p* < 0.05.

**Table 2 ijerph-19-09352-t002:** Comparison of the proportions between two groups (LCPU and HCPU).

Physical and Mental Health Variables	Low Cell Phone Users (LCPU) N = 202	High Cell Phone Users (HCPU) N = 198	Test-Statistic	*p*-Value
**Eyestrain**			191.11	0.021 ***
**Yes**	22 (10.5)	65 (33.5)		
**No**	46 (22.5)	54 (27.0)		
**Sometimes**	134 (67.0)	79 (39.5)		
**Neck Pain**			175.23	0.000 ***
**Yes**	25(12.5)	110 (55.0)		
**No**	18(8.5)	10 (6.0)		
**Sometimes**	159(79.0)	78 (39.0)		
**Weight gain**			187.14	0.000 ***
**Yes**	19 (0.09)	72 (36.0)		
**No**	24 (11.8)	60 (30.3)		
**Sometimes**	159 (78.7)	66 (33.0)		
**Back pain**			217.12	0.037 ***
**Yes**	43 (21.5)	54 (27.0)		
**No**	5 (0.2)	34 (17.0)		
**Sometimes**	154 (76.5)	110 (56.0)		
**Depression**			87.883	0.000 ***
**Low depression level (<median)**	102 (50.5)	40 (20.2)		
**High depression level (>median)**	100 (49.5)	158 (79.7)		
**Loneliness**			16.935	0.003 ***
**Low loneliness level (<median)**	104 (51.5)	141 (71.2)		
**High loneliness level (>median)**	98 (48.5)	57 (28.8)		
**Mood disorder**			147.318	0.001 ***
**Low level of mood disorders (<median)**	182 (90.5)	62 (31.3)		
**High level of mood disorders (>median)**	20 (9.5)	136 (68.7)		

Chi-square Test of Association, *** significant if *p* < 0.05.

**Table 3 ijerph-19-09352-t003:** Odds of negative impact on personality under different physical and mental health issues.

Physical and Mental Health Issues	Coefficients (95% CI)	*p*-Value
Constant	−1.778	0.000
$\;X_{1}$ **= Eye stain (yes in HCPU group)**	3.074 *** (1.875~5.040)	0.000 < 0.05
$X_{2}\;$ **= Neck pain (yes in HCPU group)**	2.583 *** 1.606~4.153)	0.000 < 0.05
$X_{3}\;$ **= Back pain (yes in HCPU group)**	2.831 *** (1.763~4.893)	0.011 < 0.05
$X_{4}\;$ **= Weight gain (yes in HCPU group)**	2.561 *** (1.502~5.812)	0.001 < 0.05
$X_{5}\;$ **= Loneliness (yes in HCPU group)**	1.295 *** (0.811~1.411)	0.000 < 0.05
$X_{6}\;$ **= Depression (yes in HCPU group)**	2.266 *** (1.672~4.812)	0.034 < 0.05
$X_{7}\;$ **= Mood disorders (yes in HCPU group)**	1.716 *** (1.065~2.763)	0.026 < 0.05

*** *p* < 0.05.

## Data Availability

The data in the study are not publicly available to protect the privacy of the participants.

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
