# Peer review of "The Relationship between Cellphone Usage on the Physical and Mental Wellbeing of University Students: A Cross-Sectional Study"

_ijerph, 2022, doi:10.3390/ijerph19159352_

Round 1

Reviewer 1 Report

Introduction

·        The concentration in the use and possession of mobile phones cannot be generalized, because there are entire countries where they are hardly used, or large areas within countries where they are extensibility used.

·        The study of the general mobile phone usage is being mixed with the specific usage; measures in time are also mixed with measures in behavior. Using the mobile for too long is clearly negative for health, like any other habit, even if it may seem healthy.

Method

·        How or to what extent is the subjectivity of measuring mental health controlled by means of self-perception tests? Wouldn't a Wong and Law (WLEIS) type test have been better?

·        What is the purpose of collecting information with personal data? A database must be available to the scientific field, then including personal data does not sound right. 

Discussion

·        There are several limitations not covered, please review.

·        Is the suggestion here to practice more regular physical activity? Is a study necessary for reaching that implication?  It seems too obvious.  

Conclusion

Need further elaboration. Now, very short and insufficient

Author Response

Peer review comments- reviewer 1

Response of author

The concentration in the use and possession of mobile phones cannot be generalized, because there are entire countries where they are hardly used, or large areas within countries where they are extensibility used.

·        The study of the general mobile phone usage is being mixed with the specific usage; measures in time are also mixed with measures in behavior. Using the mobile for too long is clearly negative for health, like any other habit, even if it may seem healthy.

Many thanks for the reviewer’s comments. I agree with the reviewer that mobile phones cannot be generalized and it also has been mentioned in the discussion section of the study that the results of this study are based upon the data collected from the response of students selected from a university of Pakistan hence limited generalizability. To address this limitation, it is proposed to have larger multi-centric studies to determine the causality effect. 

How or to what extent is the subjectivity of measuring mental health-controlled utilizing self-perception tests? Wouldn't a Wong and Law (WLEIS) type test have been better?

·        What is the purpose of collecting information with personal data? A database must be available to the scientific field, then including personal data does not sound right

The authors agree that measuring mental health by Wong and Law (WLEIS) test for measuring mental health variables. The authors also utilized authentic scales to measure the mental health variables. the details have been added in the questionnaire design section.

The study was conducted in accordance with Helsinki declaration. No personal data was collected and informed consent was obtained prior to the study. The results are presented as cumulative data where individual identity is not identified .

This was a cross-sectional study to study the impact of cellphone usage on  physical and mental health outcomes. As this research topic is novel to Pakistan, it is essential to collect basic demography data so that  we can gauge the cellphone impact on physical and mental health on specific age group , gender and population. .

The data is available upon request from the lead author if needed. Furthermore, to protect the privacy it has been mentioned in the Data Availability Statement that the data in the study are not publicly available to protect the privacy of the participants.

There are several limitations not covered, please review.

·        Is the suggestion here to practice more regular physical activity? Is a study necessary for reaching that implication?  It seems too obvious.  

 Many thanks for the reviewer’s comments. Though it could be emphasizing that physical activity is essential there are few studies which  showed the  association of cell phone usage and its impact on physical and mental health in Pakistan. Our study would provide stake holders to create policies and will allow policymakers to promulgate increase physical activity behavior through the available evidence.

Need further elaboration. Now, very short and insufficient

Many thanks for the reviewers' comment we have added the necessary points to the conclusion

Reviewer 2 Report

I applaud the authors for conducting an interesting article with much potential but it will need minor revisions in order to be published.

Abstract: In the results section, you should include the statistical indexes.

Introduction: Please explain why this study was necessary (in page 2, line 80).

Methods: Several information about the questionnaire is missing (e.g.: present examples of questions used).

Discussion: The novelty of your work must be discussed. Include recommendations for future research after the conclusion section.

Limitations: I think limitations are sub-estimated. For example, the authors believe that the use of an electronic questionnaire could be a limitation.

Minor Revisions: Page 3, line 140 – Duplicate sentence.

Author Response

Peer review comments - Reviewer 2

Author response

In the results section, you should include the statistical indexes.

In the results section, statistical results have been added

Please explain why this study was necessary (on page 2, line 80).

Many thanks for the reviewer’s comments . The point has been addressed in the aims section.

Several information about the questionnaire is missing (e.g.: present examples of questions used).

Information about questions has been addressed in the questionnaire design section

The novelty of your work must be discussed. Include recommendations for future research after the conclusion section.

The novelty of the study and recommendation has been added after the conclusion section

I think limitations are sub-estimated. For example, the authors believe that the use of an electronic questionnaire could be a limitation.

Many thanks for the comments . we have tried to address all the possible limitations  associated with cross-sectional study in the limitations section.

Page 3, line 140 – Duplicate sentence.

A duplicate sentence has been deleted